# Complete Intra-Operative Image Data Including 3D X-rays: A New Format for Surgical Papers Needed?

**DOI:** 10.3390/jcm11237039

**Published:** 2022-11-28

**Authors:** Pietro Regazzoni, Wen-Chih Liu, Jesse B. Jupiter, Alberto A. Fernandez dell’Oca

**Affiliations:** 1Department of Trauma Surgery, University Hospital Basel, 4031 Basel, Switzerland; 2Kaohsiung Medical University Hospital, Collage of Medicine, Kaohsiung Medical University, Kaohsiung 80756, Taiwan; 3Hand and Arm Center, Department of Orthopedics, Massachusetts General Hospital, Boston, MA 02114, USA; 4Department of Traumatology, British Hospital, Montevideo 11600, Uruguay; 5Residency Program in Traumatology and Orthopedics, University of Montevideo, Montevideo 11600, Uruguay

**Keywords:** intra-operative 3D X-ray, intra-operative 3D CT, artificial intelligence, evidence-based surgery, distal radius fracture

## Abstract

Intra-operative 3D X-rays have been confirmed to decrease revision rates and improve optimal screw placement in complex fractures of the distal radius. Compared with traditional surgical publications, another advantage of whole intraoperative clinical imaging can be presented in electronic databases, e.g., the ICUC working group, through a link without size limitation. The detail of complete intra-operative image dataset includes essential technical details which can be analyzed secondarily for costs and complications, considering the technical performance bias. Furthermore, the new format complies with reading/learning preferences of young surgeons and allows secondary work-up by artificial intelligence. Intra-operative 3D X-ray is a new approach for better surgical outcomes, economic benefit, and educational purposes.

A publication by Halvachizadeh et al. [1] has confirmed the advantage of using intra-operative 3D imaging for complex fractures of the distal radius by enabling a decrease in revision rates and improved optimal screw placement without increasing duration of surgery. Previous papers have also shown the economic advantages of intra-operative 3D X-rays [2,3]. A further advantage—for learning—would result from allowing access not only to complete radiological but also clinical imaging, as proposed by the ICUC working group (www.icuc.net) [4]. Electronic publishing easily allows the management of the great amount of data resulting from the implementation of such a concept. Access to complete data would also positively influence quality control efforts by allowing the confrontation of results obtained from different sources under repeated, comparable conditions [5].

In conventional surgical publications—even following very precisely formulated research protocols for the highest evidence standard like RCT—essential data, e.g., reduction maneuvers in fracture treatment, are not accessible to the readers. Skills aspects are essential for the outcome, can be measured and correlated with complications and costs [6], but cannot be analyzed secondarily without complete intra-operative image data. The inter-operator variance in technical performance is a proven reality and inevitably produces a “technical performance bias” [7]. In simple procedure, a 3D X-ray may not be necessarily needed; however, the bias is particularly important for complex surgical procedures. This seems to be a fundamental problem of surgical trials and a fundamental obstacle to sound “evidence-based surgery”. Efforts should be made to produce evidence levels similar to those well documented in clinical trials for new drug development.

Size limitations have hitherto led to the avoidance of complete intra-operative image documentations being shown in conventional surgical papers. Electronic formats have changed the situation. A short main text can show representative images, whereas the total data set can then be made accessible through a link. Such a new format also complies with reading/learning preferences of young surgeons and allows secondary work-up by artificial intelligence (AI) in highly efficient neuromorphic learning system [8,9]. Complete intra-operative image data, as proposed and realized by the ICUC working group [10,11], also allows the inclusion of essential technical details once novel surgical techniques are introduced to general public, i.e., when “non-early users” produce “real-world data”. It is well known that certain complications appear in a second use phase, often due to small deviations from the technique presented and used by inventors and early users, when the public starts to use the new method.

In conclusion, 3D intra-operative imaging is a beneficial addition rather than a requirement. To reach evidence levels comparable to drug development process, surgical trials need to adopt documentation methods including also clinical, intra-operative images. Secondary analysis of such data allows the measurement of surgical skills, avoidance of technical performance biases, and improve homogeneity of trial groups. New electronic, open-source formats allow access to the complete dataset by links, avoiding the size limitations of conventional publication formats, but managing quality control and allowing post-publication data mining.

## Data Availability

The complete intra-operative image data mentioned in this study are openly available at https://www.icuc.net.

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
