# Peer review of "Complete Intra-Operative Image Data Including 3D X-rays: A New Format for Surgical Papers Needed?"

_jcm, 2022, doi:10.3390/jcm11237039_

Round 1

Reviewer 1 Report

The authors present a viewpoint on 3D X-rays.

l. 33: instead of "same", please use "comparable".

The complex procedures are not usually the ones with a lot of RCTs covering these. In more easy approaches, a 3D X-ray may not be necessarily needed.

Likewise, internal medicine lacks reproducibility since technical procedures are described insufficiently.

Post-hoc AI work-up may be limited due to the size of these images and RAM-capacity.

Intraoperative images should not become a requirement, but rather a beneficial addition.

Author Response

l. 33: instead of "same", please use "comparable".
Thank you for the suggestions. We have made changes based on your suggestion.

The complex procedures are not usually the ones with a lot of RCTs covering these. In more easy approaches, a 3D X-ray may not be necessarily needed.

Thank you for the suggestions. We have made changes based on your suggestion. Please refer to Line 42-43.

Likewise, internal medicine lacks reproducibility since technical procedures are described insufficiently.

Thank you for the suggestions. To be clear, we substituted the internal medicine with clinical trials for new drug development. Please refer to Line 54-55.

Post-hoc AI work-up may be limited due to the size of these images and RAM-capacity.

Thank you for the suggestions. We supposed with the development of highly efficient neuromorphic learning system, machine learning will become more feasible in analyzing surgical images, as well as surgical videos.

Intraoperative images should not become a requirement, but rather a beneficial addition.

Thank you for the suggestions. We have made changes based on your suggestion in Lin 63-64.

Reviewer 2 Report

Detailed intra-operative radiographic images are sometimes essential for documentation and learning curve. Images with no size limitation are the way forward. Superior surgical outcomes are anticipated with a proper documentation of complete data and images. The practice of analysing huge amount of databases to produce new information is something well worth the effort.

Author Response

Detailed intra-operative radiographic images are sometimes essential for documentation and learning curve. Images with no size limitation are the way forward. Superior surgical outcomes are anticipated with a proper documentation of complete data and images. The practice of analysing huge amount of databases to produce new information is something well worth the effort.

Thank you for the remarks.

Reviewer 3 Report

Thank you very much for given me the opportunity to review the viewpoint article “Complete intra-operative image data including 3D X-rays: 2 A new format for surgical papers needed?”

In general, it is a well written article on the very interesting topic of intraoperative 3D imaging and the use of the collected data. Besides navigation in pelvic trauma, we use intraoperative CT imaging in distal radius fractures, intraarticular fractures and confirmation of the correct distal fibula position after fixation of a high weber c fracture.

  I have nothing to add but maybe a few thoughts you might consider:

 This (big) data might also be used in analyzing fracture morphologies which may result in an opt-

malization of implants (form and screw configuration) to capture the most common fracture patterns (helical plates are a good example)

 As far as I know no additional ethic consent is necessary if complete anonymized CT images are used.  

 Let`s hope that this article will initiate a coordinated data collection and distribution of intraoperative 3d images.

Author Response

Thank you very much for given me the opportunity to review the viewpoint article “Complete intra-operative image data including 3D X-rays: A new format for surgical papers needed?”

In general, it is a well written article on the very interesting topic of intraoperative 3D imaging and the use of the collected data. Besides navigation in pelvic trauma, we use intraoperative CT imaging in distal radius fractures, intraarticular fractures and confirmation of the correct distal fibula position after fixation of a high weber c fracture. 

I have nothing to add but maybe a few thoughts you might consider: 

This (big) data might also be used in analyzing fracture morphologies which may result in an optimalization of implants (form and screw configuration) to capture the most common fracture patterns (helical plates are a good example)

As far as I know no additional ethic consent is necessary if complete anonymized CT images are used.

Let’s hope that this article will initiate a coordinated data collection and distribution of intraoperative 3D images.

Thank you for the remarks. We appreciate your thoughts and will put them into our further clinical investigation of the application of intraoperative 3D images.